# Extracting Small Subgraphs in Road Networks

## ABSTRACT

Online navigation platforms are well optimized to solve the standard objective of minimizing travel time and typically re- quire precomputation-based architectures (such as Contraction Hierarchies and Customizable Route Planning) to do so in a fast manner. The reason for this dependence is the size of the graph that represents the road network, which is large. The need to go beyond minimizing the travel time and introduce various types of customizations has led to approaches that rely on alternative route computation or, more generally, small subgraph extraction. On a small subgraph, one can run computationally expensive algorithms at query time and compute optimal solutions for multiple routing problems. In this framework, it is critical for the subgraph to (a) be small and (b) include (near) optimal routes for a collection of customizations. This is precisely the setting that we study in this work. We design algorithms that extract a subgraph connecting designated terminals with the objective of minimizing the subgraph's size and the constraint of including near-optimal routes for a set of predefined cost functions. We provide theoretical guarantees for our algorithms and evaluate them empirically using real-world road networks.

**ACM Reference Format:**
Anonymous Author(s). 2023. Extracting Small Subgraphs in Road Networks. In *Proceedings of ACM Conference (Conference'24)*. ACM, New York, NY, USA, 9 pages. https://doi.org/10.1145/nnnnnnn.nnnnnnn

## 1 INTRODUCTION

Online navigation platforms, such as Google Maps, Waze, Apple Maps, and others, serve hundreds of millions of users on a daily basis, helping them commute to work, go on road trips, and generally navigate the world. To serve these requests, the platforms need to run billions of shortest-path queries daily. To support such volumes and to provide responses in a timely manner, these platforms rely on architectures such as Contraction Hierarchies (CH) and Customizable Route Planning (CRP) [5, 9] which enable very fast path search even in large networks. These architectures are essentially smart data structures or augmentations to the road network that are precomputed offline based on a given metric on the network's edges which typically is the travel time.

As the use of such platforms becomes more ubiquitous, the need for them to support various objectives and customizations becomes more prevalent. For example, drivers may often prefer routes that are not necessarily the fastest but have other features such as being sustainable, avoiding slow-moving traffic, being safe, or having low variance. Corporations routing fleets of trucks need to respect certain restrictions such as avoiding narrow roads, low overpasses, and sharp turns. Ride-sharing drivers might opt for routes that are safer or minimize the monetary cost. As we mentioned earlier, the known architectures rely on precomputation stages that utilize the edge lengths with respect to the cost function that will be optimized. In the presence of customizable preferences the accuracy of routes given by such precomputations suffers, the objective functions one could optimize for are large in number, and repeating the precomputation and offering a separate service for each one is not possible given the scale of the road network. Moreover, combinatorial expressions of preferences (e.g., a route without difficult maneuvers that prefers highways but avoids tolls) cause a further exponential blow-up to these requirements. In this customizable framework, it would clearly be beneficial to perform computations on a much smaller subgraph of the road network. Ideally one would like to have the capability to isolate the parts of the road network that could possibly be relevant to a particular request and then be able to perform optimization of complex objectives on them.

Another use case of this capability arises when the platform wishes to run algorithms that are more complicated than the shortest path, e.g., running algorithms for the Vehicle Routing Problem (VRP) which routes a fleet of vehicles to complete tasks such as pick-up and delivery of orders or supply chains [7]. Similarly, for tourist application variants, algorithms for orienteering and other TSP-type problems are of interest [8]. Again in this setting having a small subgraph that preserves shortest paths with respect to multiple restrictions and cost functions is extremely useful. Finally, working with a small subgraph also allows for optimizing path-dependent cost functions, e.g., cost functions that are non-linear in terms of the segment costs. Similarly, note that travel time computation is done by means of ML models, which are often more accurate when given the entire route, rather than when asked to score each segment separately and sum up their delays, giving rise to another use case of path-dependent costs.

One approach often taken by routing platforms in the effort to extract small subgraphs of the road network is to utilize algorithms that compute alternative routes [1, 2, 12, 13, 15, 17]. The idea is that if such an algorithm provides diverse and also high-quality routes, then it will recover the most reasonable ways to travel between two endpoints and, hence, will include a good route for every objective conceivable. It is clear, however, that such a process does not provide any guarantees in terms of the existence of a good path for every possible objective. In cases where the objective is to avoid certain restrictions, it is possible that a feasible path will not even be present in the subgraph. To ensure the existence of good paths for multiple objectives through the alternates approach, one would need to extract a very large number of candidate routes, resulting in unnecessarily large subgraphs. In this work, we augment the alternates approach for subgraph generation in a way that provides

*Conference'24, July 2024, Washington, DC, USA*
© 2023 Association for Computing Machinery.
ACM ISBN 978-x-xxxx-xxxx-x/YY/MM. . . $15.00
https://doi.org/10.1145/nnnnnnn.nnnnnnn

guarantees for the objective functions of interest while ensuring the subgraph is small.

Specifically, we study the following problem: We are given a graph representing the road network, a set of terminals, a collection of custom cost functions, and a large number of alternative routes between any two terminals. Each route is labeled as *feasible* or not for each cost function. Feasibility can, for example, be defined either in terms of approximating the optimal cost (i.e., feasible routes are those that have a cost at most $1+\epsilon$ times the optimal cost for the cost function under consideration, for a given parameter $\epsilon$) or in terms of ranking (i.e., the top $k$ routes are feasible, for some parameter $k$). We are asked to select a subset of the edges such that we include a feasible route for every origin, destination, and cost function triplet, and the cardinality of the set of selected edges is minimized. We provide the formal description of the model in Section 2.

## 1.1 Our Contributions

In this work, we present the first subgraph selection algorithms that provide guarantees of approximation on (a) the cost of the best-included path for multiple cost functions and (b) the size of the subgraph. The former type of guarantee is in the form of constraints that the algorithm needs to satisfy and the latter is in the form of approximation guarantees on the size of the subgraph that the algorithm achieves. We deploy techniques from submodular minimization and design greedy algorithms. We prove theoretical approximation factors for our algorithms and then put them to the test against baselines in the real road networks of New York City and Tokyo. More concretely, our results are as follows:

- We first prove NP-hardness of the subgraph extraction problem by showing equivalence to the hitting set problem.
- We then move to a warm-up case where, for every cost function, the edge costs are either 0 or $\infty$, meaning that we consider cost functions that model restrictions (e.g., closures, narrow roads for trucks, highways for scooters) and we care about preserving connectivity of the graph for all cost functions. We show that this variant of the problem remains NP-hard and then proceed to give an approximation algorithm with approximation factor logarithmic in the number of vertices and the number of cost functions.
- Going back to the general problem, we show that we can leverage an algorithm for the minimum submodular cover with submodular costs problem, to achieve an approximation that is logarithmic in the number of cost functions.
- We experimentally evaluate our algorithms and show that the approximations achieved in real road networks are in fact much better than the worst-case guarantees.
- Finally, we slightly modify our algorithms for application in a more practical setting where we are given a budget in terms of the size of the graph and wish to optimize the quality of the subgraph, in terms of the routes it includes. We evaluate our algorithm against an alternative route generator baseline.

## 1.2 Related Work

*Alternates.* The problem we consider is closely related to the literature on generating alternative routes, where the objectives are to produce a set of alternate routes that can accommodate multiple preferences and remain robust in the face of changes in network conditions. [3] employs various objective functions, including best travel time, minimum distance, road quality, scenery, and more, to compute these alternate routes. Another approach tackles the $k$-shortest path problem, as introduced by [21]. However, it is worth noting that most of the alternate routes generated through these methods tend to be minor modifications of the original path, such as exiting and immediately re-entering a highway. Consequently, these alternate routes may not perform well when subjected to different cost functions.

The most widely used methods for generating alternate routes are the via-node and plateau methods, as referenced in [1, 6]. The primary objective of these methods is to identify an intermediate stop node, known as the via-node, which allows for the establishment of the shortest path from the source to the destination. In general, most of the alternate route generation approaches take a straightforward and natural route, which involves including good routes for multiple objectives without considering the specific cost functions [1, 2, 12, 13, 15, 17]. This approach serves as the baseline we compare against in the experimental section.

*Minimum Submodular Cover with Submodular Cost.* As we demonstrate in Section 3.1, our problem is equivalent, in terms of approximation, to the Hitting Set Problem, which, in turn, is equivalent to the set cover problem. Some of our analysis also resembles the analysis of the performance guarantees of the greedy algorithm in the set cover problem [16]. The general version of our problem can be viewed as a constrained submodular minimization problem. In particular, it can be conceptualized as a minimum submodular cover with submodular cost problem [19, 20]. Other variations of the problem discussed in the literature include submodular knapsack constraints [10], generalized submodular cover [18], and submodular cost submodular cover with an approximate oracle [4].

## 2 PRELIMINARIES

In this section, we present the formal definition of our problem. We first introduce the notation needed for defining our problem and then we recall some established notations that we use in our analysis. Consider a graph denoted as $G = (V, E)$, where $V$ represents the set of vertices, and $E$ is the set of edges. A *path*, denoted as $p$, is defined as a set of edges that connects specific pairs of vertices. We use $E(p)$ to denote the set of edges in path $p$ and for a set of paths, $X$, we define $E(X) = \cup_{p \in X} E(p)$. We call a pair of vertices a *point-of-interest* (POI) pair. We consider $k$ different constraints that for each POI pair determine the set of feasible paths and use $[k] = \{1, 2, \cdots, k\}$ to denote the set of feasible constraints. (Recall that the natural interpretation of these constraints is that they express whether a route is approximately optimal for each one of $k$ cost functions.) We use $i$ to index POI pairs and $j$ to index a constraint. Following this notation, we use $P_{ij}$ to denote the set of feasible paths for POI pair $i$ and constraint $j$ and $P$ to denote the union of all these path sets. We formally define the SubgraphExtraction problem as follows.

DEFINITION 1 (SubgraphExtraction problem). *Given a graph $G = (V, E)$, a set $I$ of POI pairs, and $k$ constraints along with feasibility path sets $P = \cup_{i \in I, j \in [k]} P_{ij}$, the objective is to find a path set $Q$ from*

*P* that minimizes the set of selected edges, i.e.,

$$Q = \underset{X \subseteq P}{\arg\min}\{|E(X)| : X \cap P_{ij} \neq \emptyset \text{ for all } i \in I \text{ and } j \in [k]\}.$$

In the remainder of this section, we introduce the notation use in our analysis.

**Graph Theory.** Given a graph $G = (V, E)$, we call a proper non-empty set S, $\emptyset \subset S \subset V$, a *cut* of the graph. We the set of edges with exactly one endpoint in $S$ form the *boundary* of $S$ and are denoted by $\delta(S)$. A set of vertices in a graph that are linked to each other by paths form a *connected* component of a graph and a graph is *connected* if it has one connected component.

**Approximation Algorithms** We show that SubgraphExtraction problem is NP-hard (see Theorem 2), so we present an algorithm that finds a near-optimal solution. Given an instance of the problem, $\mathcal{P} = (G, I, [k], P)$, let $\mathsf{OPT}(\mathcal{P})$ be the optimal solution. An algorithm takes an instance as input and outputs a subset of the edges, i.e., $\mathsf{ALG} : \mathcal{P} \rightarrow Q$. We say an algorithm is $\alpha$-approximate to the optimal solution if

$$\max_{\mathcal{P}} \left\{ \frac{|\mathsf{ALG}(\mathcal{P})|}{|\mathsf{OPT}(\mathcal{P})|} \right\} \leq \alpha$$

**Submodular Function.** A *submodular* function, is a set function with diminishing marginal value gain over larger sets, i.e., adding an element to a set always increases the function value by at least as much as adding the same element to a larger set. More formally, given a finite set $U$, a set function $f : 2^U \rightarrow \mathbb{R}$, where $2^U$ denotes the power set of $U$, is a submodular function if for every $X \subset Y \subset U$, and every $e \in U \setminus Y$:

$$f(X \cup \{e\}) - f(X) \geq f(Y \cup \{e\}) - f(Y).$$

The marginal value of a set $Y$ with respect to $X \subseteq U$ is defined by

$$\Delta_Y f(X) = f(X \cup Y) - f(X).$$

For the case that X is a singleton, i.e., $X = \{e\}$, we simplify the notation to $\Delta_e f(Y)$. We say a function $f$ is increasing if $\Delta_e f(X) \geq 0$ for all $X \subseteq U$ and $e \in U \setminus X$. A submodular and increasing function f is called a polymatroid function if $f(\emptyset) = 0$.

## 3 THEORETICAL RESULTS

### 3.1 Equivalence to the Hitting Set Problem

In this section, we show that the SubgraphExtraction problem is NP-hard by presenting an approximation-preserving reduction from the HittingSet problem. In the HittingSet problem, we are given a universe $U$ of elements and a collection $\mathcal{S} = \{S_1, S_2, \ldots, S_m\}$ of subsets of $U$; the objective is to find a subset $H \subseteq U$ of minimum cardinality such that every subset $S_i \in \mathcal{S}$ contains at least one element from $H$. The problem is NP-hard [11], and it is approximable within $\log(|U|)$ using the greedy algorithm.

**Theorem 2.** *There is an approximation preserving reduction from the HittingSet problem to the SubgraphExtraction problem.*

**Proof.** We propose a reduction from HittingSet to SubgraphExtraction where there is a 1-1 mapping between solutions of the two instances such the size of the solution for SubgraphExtraction is exactly twice the size of the corresponding solution of

the HittingSet, therefore any approximation algorithm for SubgraphExtraction can be employed to give the same approximation for the HittingSet problem. Given an instance of the HittingSet problem, we construct an instance of the SubgraphExtraction by defining components as follows:

- $G = (V = U \cup \{\eta, \beta\}, E = \{(v, w) : v \in \{\eta, \beta\}, w \in U\}$.
- $I = \{\eta, \beta\}$.
- $m$ constraints with feasibility path set $P_j = \{(\eta, u, \beta) : u \in S_j\}$ (since there is only one POI pair, we simplify $P_{1j}$ to $P_j$).

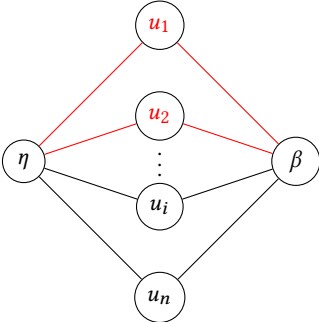

**Figure 1: An example showing the reduction from HittingSet to SubgraphExtraction , with $S_1 = \{u_1, u_2\}$. In the instance of SubgraphExtraction , we would have $P_1 = \{(\eta, u_1, \beta), (\eta, u_2, \beta)\}$ (marked in red).**

Note that any feasible solution $H$ to the HittinSet problem, can be translated to a feasible solution $Q = \cup_{u \in H}\{(\eta, u, \beta)\}$ since $H$ intersects with $S_j$, $Q$ contains at least one path from $P_j$. Note that the size of $|Q| = 2|H|$ as our paths contain exactly two edges. Similarly, any feasible solution $Q$ of SubgraphExtraction can be translated to a feasible solution $H = \cup_{(\eta,u,\beta) \in Q}\{u\}$ since $Q$ contains a path from each $P_j$, so $H$ contains at least one element from $S_j$. This argument shows that any approximation of the SubgraphExtraction problem translates to a solution with the same approximation for the HittingSet . □

**Corollary 3.** *SubgraphExtraction is NP-hard.*

### 3.2 The Common Spanning Subgraph Problem

In this section, we consider a special case of the problem where (i) all pairs of vertices are POI pairs, i.e., $I = \binom{V}{2}$, (ii) for each constraint $j$, we have an input *feasibility* graph $G_j = (V, E_j)$ where any path in this graph is a feasible path. We assume that all these graphs are connected so $P_{ij} \neq \emptyset$ for any POI pair $i$ and constraint $j$. In this special case, we are looking for a small subgraph that induces a spanning subgraph in each feasibility graph. We henceforth call this special case the CommonSpanningSubgraph problem. We restate the definition of SubgraphExtraction for this special case:

**Definition 4 (CommonSpanningSubgraph Problem).** *Given a graph $G = (V, E)$, and connected constraint graphs $G_j = (V, E_j)$, the objective is to find a subset of edges $Q \subseteq E$ with minimum size such that $Q$ yields a connected graph for each constraint, i.e., all pairs are connected:*

$$Q = \underset{X \subseteq E}{\arg\min}\{|X| : X \cap E_j \text{ is connected for each } j \in [k]\}.$$

We first demonstrate that even this special case of the problem is NP-hard. We then propose a greedy algorithm that achieves an $O(\log n \log k)$-approximation, where $n$ is the number of vertices, and $k$ is the number of different constraints (feasibility graphs). All the missing proofs are deferred to Appendix A.

LEMMA 5. *The CommonSpanningSubgraph problem is NP-hard.*

In the remainder of this section, we focus on the CONNECTIV-ITYGREEDY algorithm and its analysis. The high-level idea is to iteratively select an edge that merges the largest number of connected components over different feasibility graphs. More formally, we start with an empty set of edges, $Q$. In each iteration, $Q \cap E_j$ induces a subgraph of the feasibility graph $G_j$ with some connected components. For each edge $e$, we compute a score that is the number of constraints for which the addition of $e$ connects different connected components. We augment $Q$ with the edge with the highest score. The algorithm stops when $Q$ induces a connected subgraph for all constraints. More formally, our algorithm is as follows and a sample run is depicted in Figure 2

---

**Algorithm 1:** CONNECTIVITYGREEDY

1 Initialize $C \leftarrow [k], Q \leftarrow \emptyset$
2 For each $e \in E$, initialize $F_e \leftarrow \{i : e \in E_i\}$ and $s_e \leftarrow |F_e|$
3 **while** $C$ *is not empty* **do**
4     $e = \arg\max_{e \in E} s_e$
5     $Q \leftarrow Q \cup \{e\}$
6     **for** $i \in C$ **do**
7         **for** $e = (u, v) \in E_i$ **do**
8             **if** $u, v$ *are connected in* $Q$ **then**
9                 $s_e \leftarrow s_e - 1$
10         **if** $Q \cap E_i$ *spans* $G$ **then**
11             $C \leftarrow C \setminus i$
12 **return** $Q$

---

THEOREM 6. *CONNECTIVITYGREEDY is a $O(\log n \log k)$-approximation algorithm for the CommonSpanningSubgraph problem.*

To prove this theorem, we use an argument based on a novel charging scheme. We define a function $q_j$ for each constraint $j$ which charges the cost of our solution, $|Q|$, to unique cuts (subsets of vertices). More formally, we define an auxiliary function $q_j : 2^V \rightarrow \mathbf{R}_{\geq 0}$ for each constraint $j$ following the execution of the algorithm. We start by setting all $q_j(S) = 0$ for all $j$ and all cuts $S$. In iteration $t$, let edge $e = (u, v)$ be the selected edge with score $s_e$. So there are exactly $s_e$ constraints where $e$ connects two connected components. For each such feasible constraint $j$, let $S_j^u$ and $S_j^v$ be the two distinct components containing $u$ and $v$. We set the $q_j(S) = 1/s_e$ where $S$ is the smaller connected component between $S_j^u$ and $S_j^v$ (see Figure 3 for an illustration of this scheme). Our charging scheme in fact satisfies the following.

LEMMA 7. $\sum_{j,S} q_j(S) = |Q|$.

PROOF. We first show that each function $q_j$ is well-defined. Note that after adding $e$ in iteration $t$, the two newly connected components for constraint $j$, can no longer take a non-zero value by the

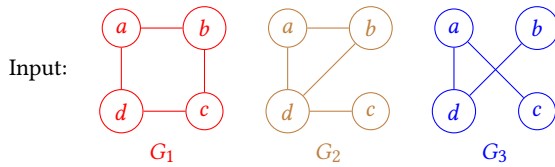

Progress of Algorithm 1

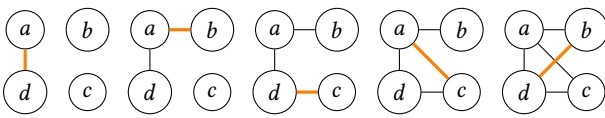

**Figure 2: An example showing the execution of Algorithm 1 on input graphs $G_1, G_2, G_3$. We break the ties by comparing the lexicographic ordering of the edge endpoints. In each iteration, the orange edge is selected. The algorithm first picks the edge $(a, d)$ as it has a score of $3$ (it is present in all feasibility graphs and connects two previously not connected vertices). In the third iteration, the algorithm picks the edge $(c, d)$ with score $2$. Note that the edge $(b, d)$ has score $1$ in the third iteration since $b$ and $d$ are already connected in $G_2$ at this point.**

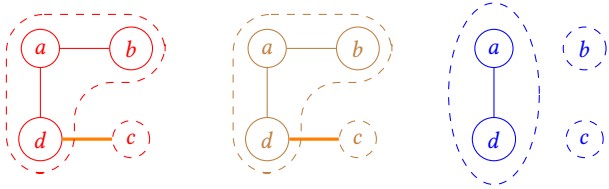

**Figure 3: An example showing how function $q_j$ assigns values. At this point, the algorithm has selected 2 edges and has picked the orange edge which has the highest score ($s_e = 2$). The connected components for each constraint are shown in dashed lines. For both constraints, the connected components are $\{a, b, d\}$ and $\{c\}$ and the smaller components gets value of $1/s_e = 1/2$, i.e., $q_1(\{c\}) = q_2(\{c\}) = 1/2$.**

function $q_j$ so there is at most one chance of getting a non-zero value by the function $q_j$ for any cut $S$. Additionally, for selected edge $e$ with score $s_e$, exactly $s_e$ sets take value $1/s_e$ (for different $s_e$ auxiliary functions) which sums up to 1. So the overall sum over all functions and sets is equal to $|Q|$. □

Now that we established our charging scheme which exactly distributes the value $|Q|$ among different unique sets over different auxiliary functions, we need to connect this scheme to the cost of the optimal solution. To this end, we rely on the greedy logic that, in each step, among all the edges in the boundary of connected components, picks the edge with the most presence. For each edge, we upper-bound the contribution of all auxiliary functions for all the cuts that have this edge in their boundary.

LEMMA 8. $\sum_{j,S:\delta_j(S) \ni e} q_j(S) = O(\log n \log k)$ *where $n$ is the number of vertices and $k$ is the number of constraints.*

Proof. For this proof, we rely on the following claim, the proof of which we defer to Appendix A.

Claim 9. *Given any edge $e = (u, v) \in \cup E_j$, let $j$ be the $a$-th constraint with respect to which $u, v$ become connected (break ties arbitrarily). For all the cut $S \in V$ such that $e \in \delta_j(S)$,*

$$q_j(S) \le \frac{1}{f_e - (a-1)}.$$

By the above claim, we establish an individual bound in terms of the $q$-value for cuts for each constraint $j$. Now, we bound the number of positive $q$-values for each constraint $j$ for some edge $e = (u, v)$. Note that the connected components around $u$ and $v$ grow as the algorithm adds edges and edge $e$ is in the boundary of all these connected components until they are merged. Now for a set $S$ in round $t$ that has $e$ in its boundary, if $q_j(S)$ is non-zero, then the size of the next connected component containing $S$ is at least twice the size of $S$ (since $S$ gets merged with a bigger set). Since this size doubles every time, there are at most $\log n$ such sets for each constraint and fixed edge $e$. Combining this with Claim 9, we obtain:

$$\sum_{j, S: \delta_i(S) \ni e} q_{\delta_i(S)} = \sum_{j \in [k]} \sum_{S} q_{\delta_i(S)}$$

$$\le \sum_{a \in [k]} \log n \cdot \frac{1}{f_e - (a-1)}$$

$$= O(\log n \cdot \log k).$$

where the last inequality follows by the bound on the harmonic series, i.e., $\sum_{i=1}^{k} 1/i = O(\log k)$. □

We are now ready to prove the main theorem of the section

Proof for Theorem 6. Consider the size of the greedy solution $|Q|$, by Lemma 7, we have:

$$|Q| = \sum_{j \in [k]} \sum_{S} q_{\delta_j(S)})$$

$$\le \sum_{e \in \mathsf{OPT}} \sum_{j, S: \delta_j(S) \ni e} q_{\delta_j(S)}) \quad \text{(since OPT spans } G \text{ w.r.t any } j)$$

$$\le \sum_{e \in \mathsf{OPT}} O(\log n \ln k) \quad \text{(by Lemma 8)}$$

$$= |\mathsf{OPT}| O(\log n \ln k). \quad □$$

## 3.3 The Subgraph Extraction Problem

We now consider the general path selection problem where we are given $P_{ij}$ for each POI pair $i \in I$ and each constraint $j \in [k]$. The objective is to output a small subgraph that includes at least one path from each of the $P_{ij}$ set. To this end, we argue that our problem can be thought of as an instantiation of the *minimum submodular cover with submodular cost* problem.

Definition 10 (Minimum Submodular Cover with Submodular Cost problem [19]). *Consider a polymatroid function $f$ and a finite set $U$, a set $X \subseteq U$ is said to be a submodular cover of $(U, f)$ if $f(X) = f(U)$. Given two submodular functions $f, g$, the minimum submodular cover with submodular cost problem is to find subset $X$ such that*

$$\min\{g(X) : f(X) = f(U), X \subseteq U\}.$$

We first define a set function $f : 2^{\mathcal{P}} \to \mathbb{R}$ as follows:

$$f(X) = |\{P_{i,j} : P_{i,j} \cap X \ne \emptyset\}|,$$

in other words, $f$ counts how many of the $P_{ij}$ set are "hit" by a set of path $X \subseteq \mathcal{P}$.

Lemma 11. *$f$ is a polymatroid function.*

Proof. First, note that $f(\emptyset) = 0$. Consider any two path sets $X$ and $Y$ such that $X \subset Y \subseteq \mathcal{P}$, and let $P(X) = P_{ij} : P_{ij} \cap X \ne \emptyset$ and $P(Y) = P_{ij} : P_{ij} \cap Y \ne \emptyset$. We first observe that $P(X) \subseteq P(Y)$. Now, consider any path $p \in \mathcal{P} \setminus Y$. Let $\Delta_p(Y)$ be the number of extra $p_{ij}$ hit by $Y \cup p$ compared to $P(Y)$. Since $P(X) \subseteq P(Y)$, we have $\Delta_p(X) \ge \Delta_p(Y)$. Note that $\Delta_p(Y) = \Delta_p f(Y)$ by the definition of $f$. Therefore, we have:

$$f(X \cup \{p\}) - f(X) \ge f(Y \cup \{p\}) - f(Y),$$

which makes $f$ a submodular function. Additionally, since including more paths in $Q$ always weakly increases the number of $P_{ij}$ hit by $Q$, $f$ is an increasing function. Since the function $f$ is submodular, increasing, and $f(\emptyset) = 0$, $f$ is, therefore, a polymatroid function. □

Recall that for a set X of paths, we use $E(X)$ to denote the edges. Let function $g$ on subset of paths $\mathcal{P}$ be defined as $g(X) = |E(X)|$. It is easy to see that this $g$ is submodualar. Note that since our objective is to minimize the size of the subgraph while hitting all of $P_{ij}$ sets, we can write our problem as follows:

$$\min\{g(Q) : f(Q) = f(\mathcal{P}), Q \subseteq \mathcal{P}\}.$$

By Definition 10, our problem is a special case of the minimum submodular cover with submodular cost problem. We tailor the greedy algorithm for this problem to our problem (see Algorithm 2) and then can use the following result to get the approximation of the greedy algorithm.

Theorem 12 ([19]). *Greedy algorithm is a $\rho H(\gamma)$-approximation algorithm for minimum submodular cover with submodular cost problems with submodular functions $f, g$ where $\gamma = \max_{x \in U} f(x)$ and $\rho$ is the curvature of the submodular cost $g$ formally defined as:*

$$\rho = \min_{S:\text{min-cost cover}} \frac{\sum_{x \in S} g(x)}{g(S)}.$$

---

**Algorithm 2:** GeneralGreedy

1 $Q \leftarrow \emptyset$;
2 **while** $\exists p \in \mathcal{P}$ such that $\Delta_p f(X) > 0$ **do**
3      $p = \arg\max \frac{\Delta_p f(X)}{g(p)}$;
4      $Q \leftarrow Q \cup \{p\}$;
5 **return** $X$

---

In our setting, since each path $p$ only serves one POI pair, we have $\gamma = \max_{p \in \mathcal{P}} f(p)$ is bounded by $k$. Combining with the analysis in [19], we get the following approximation guarantee of GeneralGreedy.

Corollary 13. *GeneralGreedy is a $\rho H(k)$-approximation algorithm for the SubgraphExtraction problem, where*

$$\rho = \min_{\mathsf{OPT}(\mathcal{P})} \frac{\sum_{p \in \mathsf{OPT}(\mathcal{P})} g(p)}{g(\mathsf{OPT}(\mathcal{P}))}.$$

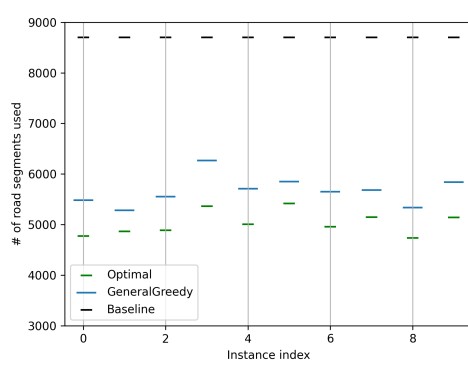

**Figure 4: The performance of GENERALGREEDY vs the optimal solution and the penalty method baseline (with 10 alternates per origin-destination pair) in randomly generated instances**

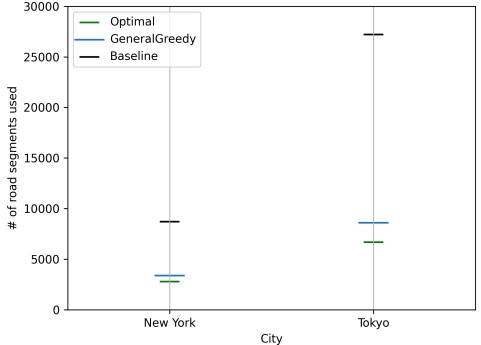

**Figure 5: The performance of GENERALGREEDY vs the optimal solution and the penalty method baseline (with 10 alternates per origin-destination pair) in actual instances**

One can interpret the $\rho$ as the level of "double counting" of edges in the optimal solution. The more edges shared among the paths in the optimal solution, the higher the $\rho$.

## 4 EXPERIMENTS

In this section, we evaluate variants of our algorithm experimentally. Given that our work is the first to consider explicitly optimizing the subgraph for given objectives, the only baselines that we can compare against from prior work are the alternate route subgraph generation algorithms that are oblivious to the objectives. We report the results against the *penalty method* [12], which is known to be a high-quality solution for the alternate route subgraph generation problem.

## 4.1 Road Network and Cost Functions

We evaluate our algorithms on two real-world road networks, specifically, the road networks of New York City and Tokyo. We extract the graphs for these cities from OpenStreetMap [14]. In each one of

the cities, we select 8 random points of interest that are the terminals that we wish to connect, yielding, in turn, 64 origin-destination pairs in $I$. In a real application, these could represent central hubs or the places that a particular user frequently visits.

Our feasible path sets $P_{ij}$ are defined through four cost functions, for which we wish to maintain near-optimal paths. These functions are the following:

- *The travel time.* We extract the travel time of a road segment by dividing the length by the speed limit as given in OpenStreetMap.
- *The length*, i.e., the total distance driven on the path.
- *A rate-card* type cost, i.e., a combination of the travel time and the distance traveled.
- *Avoid highways.* Here the base cost is the travel time and a penalty (we use the value of 100 seconds per highway road segment for our experiments) is added for each highway segment in the route. In the OpenStreetMap data, we consider any segment with 3 or more lanes to be a highway.
- *Avoid narrow roads.* Again the base cost is the travel time, but here a penalty (again we use the value of 100 seconds per narrow road segment for our experiments) is added for narrow roads, which we define as those having a single lane.

We aim to select a subset of the edges with cardinality as small as possible that contains approximately optimal paths for every pair of terminals and every cost function. The "approximate optimality" (for a single trip and cost function) here is defined in two ways, one in each of the following two sections. In the first one (the *satisfactory routes variant* of the experiment), we put a path in $P_{ij}$ if and only if the path is in the top four paths between the endpoints of $i$ for cost function $j$. The second variant of the experiment (the *accuracy vs graph size variant*) measures approximate optimality using the cost functions themselves by means of the *accuracy* metric:

DEFINITION 14 (ACCURACY). *We say the accuracy of trip* $p \in \mathcal{P}_i$ *with respect to some cost function* $c^j$ *is*

$$a^j(p) = \frac{\min_{\ell \in \mathcal{P}_i} c^j(\ell)}{c^j(p)}.$$

We note that $0 < a^j(p) \le 1$. We define the metric for the entire subgraph as the minimum of the maximum accuracy of any cost functions and POI pairs. In other words, the accuracy of a subgraph is the minimum accuracy of the best trip included for each cost function with respect to each POI pair. Formally:

DEFINITION 15 (ACCURACY-LEVEL). *Given a subset of edges* $Q$, *we say the accuracy level of the subset is*

$$\min_{i,j} \max_{p \in Q} a^j(p)$$

The trade-off of interest is then the one between the size of the subgraph in terms of the number of edges and its accuracy level.

## 4.2 Satisfactory Routes Variant

In the first round of experiments, we evaluate the algorithm proposed in Section 3. For each origin, destination, and cost function triplet, some of the routes are designated as *satisfactory* and the subgraph needs to include at least one such path per triplet. In general

applications the designation can be done in any way, e.g., requesting that the path is among the top $k$ paths for the cost function under consideration, or that the path satisfies some approximate optimality threshold. For our experiment, we deem a path as satisfactory if it falls within the top four routes among the ones for the corresponding POI pair and cost function. Then we request that the subgraph includes at least one of the top four paths for every origin, destination, and cost function, and seek to minimize the number of included edges.

We first test GENERALGREEDY on synthetic instances. To generate multiple instances, we use synthetic cost functions (noting that each city represents a single instance for natural cost functions). We randomly label some of the paths as satisfactory for each of the cost functions, with specifically four paths labeled as such for each origin, destination, and cost function triplet.

In Figure 4, we present the performance of GENERALGREEDY over the random instances. We also plot the optimal solution, extracted using a (very slow and non-practical) linear programming-based algorithm, alongside the penalty method baseline that extracts 10 alternates for each origin-destination pair. It is worth noting that the performance of GENERALGREEDY exceeds the theoretical guarantees. On average, the solution output by GENERALGREEDY is only 1.12 times the size of the optimal graph. Additionally, as observed, by keeping, on average, only 65% of the initial subgraph, we can ensure that for any source, destination, and cost functions, at least one satisfactory trip is included.

We also test GENERALGREEDY on real-world instances with actual cost functions discussed in Section 4.1. Similar to the synthetic instances, we consider a path satisfactory if it ranks among the top four routes for the same pair of points of interest (POI) with respect to the corresponding cost functions. We then require that the subgraph includes at least one of the top four paths for every origin, destination, and cost function while seeking to minimize the number of included edges.

In Figure 5, we present the performance of GENERALGREEDY for New York and Tokyo. For New York, where the subgraph size of the penalty method alternates baseline is 8, 699, the subgraph output by GENERALGREEDY contains 3, 364 edges, while the optimal subgraph has a size of 2, 788. For Tokyo, the baseline graph contains 27, 207 edges, the GENERALGREEDY subgraph has 8, 583 edges, while the optimal subgraph contains 6, 661 edges. We note that the full graph sizes for the two cities are in the hundreds of thousands. We observe that our algorithm recovers most of the headroom towards the optimal solutions.

## 4.3 Accuracy vs Graph Size Variant

Now we turn to a variant of the experiment that we deem as more practically relevant: Suppose we are given a budget on the size of the subgraph and we would like to ensure approximate optimality of the included paths as much as possible. In this regard, we can keep adding edges or paths to our subgraph until we have reached the available budget. It is then interesting to plot what the different algorithms can achieve in terms of route quality as a function of the size of the subgraph that they are allowed to construct. We will again compare the performance of our greedy algorithm (after we slightly modify it to fit the new setting) against the penalty method

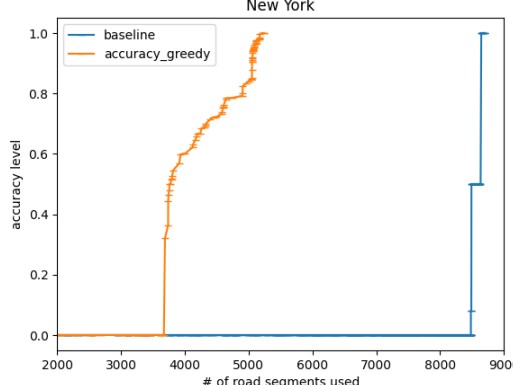

**Figure 6: more refined accuracy size trade-off for New York**

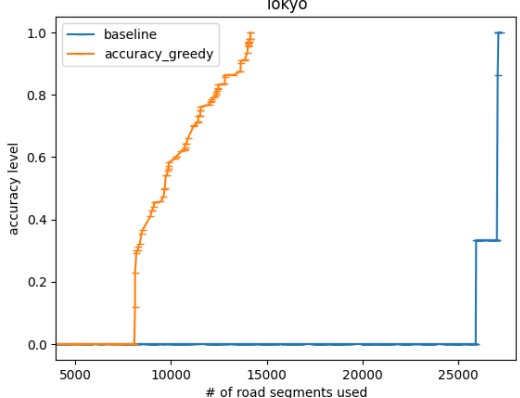

**Figure 7: more refined accuracy size trade-off for Tokyo**

alternates baseline. To extract the tradeoff for the baseline, we let the number of alternates extracted per origin-destination pair vary. As we will see, the penalty method requires around 10 alternates to perform well, which is also the reason why 10 was selected as the de facto value in the previous section's experiment.

Note that the GENERALGREEDY algorithm terminates once it has included at least one feasible trip for each cost function $j$ with respect to each POI pair $i$. To fully utilize the concept of accuracy and provide more possible trade-off options, we modify GENERALGREEDY as follows: we again start with an empty set $Q$. At any point, the algorithm maintains an accuracy table that keeps track of the best accuracy for each cost function in each POI pair. In each iteration, the algorithm identifies the POI pair $i$ that currently has the minimum accuracy in the accuracy table and selects a trip $p \in P_i$, maximizing the difference between the minimum accuracy of the POI before and after adding $p$. We continue this process until we exceed the budget for the included edges. The pseudocode of the modified algorithm, called MODIFIEDGREEDY, is provided in Algorithm 3.

Figures 6 and 7 plot the performance of our algorithm and the penalty method baseline against the size of the subgraph. We observe that our algorithm achieves full accuracy using considerably

---

**Algorithm 3:** MODIFIEDGREEDY

---

**1** Input: edge budget $B$

**2** Initialize: $Q \leftarrow \emptyset$;

**3** **while** $|\cup_{p \in Q} e(p)| \leq B$ **do**

**4**      Compute the accuracy table $A(Q)$

**5**      $a_{i^*, j^*} \leftarrow \underset{i,j}{\arg\min}\, A_{i,j}$

**6**      $p \leftarrow \underset{p \in P_{i^*}}{\arg\max}\, \min(A(Q \cup p)[i:]) - \min(A(Q)[i:])$

**7**      $Q \leftarrow Q \cup p$

**8** **return** $Q$

---

fewer road segments. For both New York and Tokyo, the experiments suggest that our method reaches maximum accuracy using only about half of the road segments that the baseline uses. In fact, we can see that in order to get any positive accuracy (let alone one that approaches 1), the baseline must already use a lot more edges than our algorithm needs to converge to full accuracy. This considerable discrepancy highlights the efficiency of our procedure.

## 5 CONCLUSION

In this work, our focus lies in designing algorithms for the efficient extraction of subgraphs that not only minimize size but also incorporate near-optimal routes for a predefined set of cost functions. Initially, we established the NP-hardness of the subgraph extraction problem by demonstrating its equivalence to the hitting set problem. Subsequently, we introduced a logarithmic approximation algorithm tailored for cases where edge costs are limited to binary values. We then applied a minimum submodular cover with submodular costs algorithm, achieving an approximation that scales logarithmically with the number of constraints induced by the cost functions. Our empirical evaluation on real road networks confirmed the strong performance of our algorithms, surpassing the worst-case guarantees and demonstrating their practical applicability.

Furthermore, to enhance the practicality of our proposed solutions, we modified the algorithm to provide a more refined trade-off between the subgraph size and the minimum accuracy for any cost function. Once again, we demonstrated that our approach significantly surpasses the existing baseline by achieving an optimal or near-optimal accuracy level using a substantially smaller subgraph.

Overall, We initialize the study on subgraph extraction in facilitating efficient and adaptable navigation systems that can cater to a diverse range of user preferences and constraints, thereby enhancing the customizability of online navigation platforms.

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

# A MISSING PROOFS FROM SECTION 3.2

Lemma 5. *The* CommonSpanningSubgraph *problem is NP-hard.*

Proof. It is known that the Hitting set problem is NP-hard. We now propose the following reduction from HittingSet to CommonSpanningTree as follows: consider an instance of the hitting set problem, with $\{S_1, S_2, \ldots, S_m\}$ and a number $b$. We construct an instance in our problem as follows:

- Let graph $G = (V = U \cup \{\eta, \beta\}, E = \{(v, w) : v \in \{\eta, \beta\}, w \in U\})$.
- Define $m$ constraints where $E_i = \{(w, \eta) : a \in S_i\} \cup \{(a, \beta) : a \in \cup_i S_i\}$ for all $i$. In other words, $E_i$ contains $(w, \beta)$ for all $w$ and $(w, \eta)$ only for $w \in S_i$.

Assume for contradiction, that there exist an polynomial time algorithm that can find the subset of edges $Q$ with the smallest carnality, such that $Q \in E_i$ spans the all vertices for all $E_i$. First note that we can wlog assume $Q$ contains all the edges $(a, \beta)$[1]. We now argue that $H = \{w : (w, \eta) \in Q\}$ is the optimal solution for the HittingSet problem. Assume for contradiction that there exist a better solution $H'$ such that $H' \cap S_i \neq \emptyset$, and $|H'| < |H|$. Note that the set $Q' = \{(w, \eta) : w \in H'\} \cup \{(w, \beta) : \forall w\}$ is a valid solution the CommonSpanningTree problem, since for each $E_i$, all of the vertices $w$ is connected to $\beta$ and at lest one $w$ is connected to $\eta$ by definition that $H'$ is a solution of the HittingSet problem and the construction of $Q'$, contradicting with the assumption that $Q$ is optimal. $H$ is therefore the optimal solution for the HittingSet problem. Since the HittingSet problem is known to be NP-hard, we therefore conclude the CommonSpanningTree problem is also NP-hard.

To show that the CommonSpanningTree is also in NP, we can have the certificate be the proposed subset of edges $Q$ and we can simply check if $Q \cap E_i$ connects all the vertices for all $i$, which is doable in polynomial time. We therefore conclude that the CommonSpanningTree problem is NP-complete. □

Claim 9. *Given any edge* $e = (u, v) \in \cup E_j$, *let* $j$ *be the* $a$-th *constraint with respect to which* $u, v$ *become connected (break ties arbitrarily). For all the cut* $S \in V$ *such that* $e \in \delta_j(S)$,

$$q_j(S) \leq \frac{1}{f_e - (a - 1)}.$$

Proof. We first make the following observation based on the greedy nature of our algorithm.

Observation 16. *Consider an edge* $e = (u, v) \in \cup E_j$, *suppose at some point of the execution,* $u, v$ *is connected with respect to* $\ell$ *number of feasible constraints, then the next edge chosen by the greedy algorithm has a score at least:*

$$f_e - \ell, \tag{1}$$

*where* $f_e = |F_e|$ *is the number of* $E_j$ *that include* $e$.

Proof. By definition of $s_e$, we have at this point $s_e = f_e - \ell$. Indeed, choosing the edge $e$ would attain the ratio in (1); the ratio

of the edge chosen by the greedy algorithm can only be weakly bigger. □

Lemma follows immediately from the above observation in the case where the $(u, v)$ is connected in each feasible constraint one-by-one (with $a - 1$ replacing $\ell$ for each $a$). In general, $u, v$ might be connected in multiple feasible constraints at the same time (e.g., the greedy algorithm might actually pick $e$). In this case, the number of feasible constraints with respect to which $u, v$ is already connected is weakly less than $a - 1$, making $q_{\delta_i(S)}$ even smaller. □

---

[1]if there exist an edge $(w', \beta)$ that are not included in $Q$, then by connectivity we have $(w', \eta)$ has to be in $Q$ and we can always remove $(w', \eta)$ and add back $(w', \beta)$ without losing connectivity and this process weakly decreases the number of edges in $Q$

