# OpenReview forum: "Extracting Small Subgraphs in Road Networks"
_ACM.org/TheWebConf/2024/Conference — TheWebConf24_

### Official Review · Reviewer_451d · 2023-11-23

**Novelty:** 5
**Technical Quality:** 5

**Review:**

## Summary

The authors designs an algorithm for extracting subgraphs and provides a detailed theoretical proof. This paper is adequate in terms of algorithmic proof. Limited experimental evaluation shows that these algorithms exceed worst-case guarantees and have some applicability.

## Strengh
+ The problem studied is interesting. It addresses the complex challenge of extracting small subgraphs from large road networks for online navigation platforms.
+ Theoretical Foundation and Algorithm Design. The theoretical proof of the algorithm is adequate.
+ Clarity of the writing. The paper is well-written and the results are well-presented.

## Weakness
- Insufficient experimental analysis. While the paper includes empirical evaluations, there is a need for more comprehensive experimental data and deeper analysis.
- Layout and Organization Issues. There are some problems in the layout of the content of the paper, and there are some methodological contents arranged in the experimental part.

**Questions:**

1. There are a number of methodological contents in Session 4.3 that should be placed in the corresponding sections of the theoretical methodology.
2. Experimental data is scarce, some experiments should be added and the corresponding data should be discussed and analysed.

**Reviewer Confidence:**

2: The reviewer is willing to defend the evaluation, but it is likely that the reviewer did not understand parts of the paper

**Scope:**

2: The connection to the Web is incidental, e.g., use of Web data or API

---

### Official Review · Reviewer_YXqg · 2023-11-23

**Novelty:** 4
**Technical Quality:** 6

**Review:**

This paper tackles the significant challenge of extracting small subgraphs in road networks to optimize various objectives for online navigation platforms. The proposed algorithms focus on extracting a subgraph that connects designated terminals while aiming to minimize the subgraph's size and ensuring near-optimal routes for predefined cost functions. These algorithms are tested on real-world road networks, demonstrating promising approximation results.

Pros:
*  The paper addresses a critical issue in online navigation, offering a practical algorithmic solution.
*  The authors not only propose algorithms but also establish their theoretical basis through mathematical proofs. Associating the SubgraphExtraction problem with NP-hardness and providing a approximation guarantee for the ConnectivityGreedy algorithm enhances the accessible of the methods.
*  The empirical evaluation of the algorithms demonstrate the effectiveness of the proposed approaches.

Cons:
* Although the algorithms are theoretically sound, there may be computational complexity issues in practical applications. Especially for large networks, the efficiency and scalability of the algorithms could be challenging.
*  The paper could benefit from a clearer exposition of the problem statement and the proposed algorithms to improve reader understanding.
* The article might lack extensive testing and validation of the algorithms in actual road networks. The evaluation of the algorithms could be further improved by comparing them with other widely-known baselines in the field.

**Questions:**

### Questions:
1. The article seems to focus on road networks. How adaptable are your algorithms to other types of networks, such as social networks or biological networks?
2. Could you provide more information on any real-world testing or case studies that have been conducted?
3. How well do the proposed algorithms adapt to dynamic changes, such as road closures or new road constructions, in real-time?

**Ethics Review Description:**

n.a.

**Reviewer Confidence:**

3: The reviewer is confident but not certain that the evaluation is correct

**Scope:**

3: The work is somewhat relevant to the Web and to the track, and is of narrow interest to a sub-community

---

### Official Review · Reviewer_Y3AN · 2023-11-23

**Novelty:** 5
**Technical Quality:** 5

**Review:**

This paper presents a novel subgraph extraction algorithm under predefined feasibility constraints to achieve minimal graph size and near-optimal solutions.

**Strengths**:

1.	The writing is good, and the logic inside the derivation of the algorithm is clear.
2.	This paper presents a significant issue of increasing the efficiency of customized online navigation problem in real world, and formulates it as a provable NP-hard graph problem. The solution of this problem utilizes a greedy-based approximation algorithm with controllable error bound and can also optimize the size of the feasible set, which is theoretically sound.
3.	The experiments show that the algorithm can successfully control the size of the extracted subgraph and near-optimal solution.

**Weaknesses**:

1.	There are some notations or typos required to be clarified: What does $\delta_j(S)$ in Lemma 8 mean? Should the return item in Algorithm 2 be Q instead of X?
2.	The author should provide a more straightforward insight of the algorithm to make the reader easier to understand the mechanistic meaning of the solution. Currently, we have to read through all the theories and lemmas to understand how this works, and this paper has applied quite a lot of mathematical tools for derivation, which increases the difficulty of understanding the method.
3.	The experiments only include one baseline method for comparison. I am wondering if there are other methods that should be taken into account. Otherwise, the results seem not that persuasive in spite of the significant performance.

**Questions:**

See Weaknesses in Review

**Ethics Review Description:**

No ethical issues

**Reviewer Confidence:**

2: The reviewer is willing to defend the evaluation, but it is likely that the reviewer did not understand parts of the paper

**Scope:**

4: The work is relevant to the Web and to the track, and is of broad interest to the community

---

### Official Review · Reviewer_hkto · 2023-11-24

**Novelty:** 5
**Technical Quality:** 5

**Review:**

Summary:
This work studies the problem of subgraph extraction with pre-defined cost constraints. This paper proves the NP-hardness of the studied problem and then introduces an approximation algorithm for the special case where edge costs are binary values. Then this work utilizes an algorithm for the minimum submodular cover with submodular costs problem, aiming to attain an approximation that exhibits logarithmic scaling with the number of cost functions. The experimental results provided in the paper underscore the effectiveness of the proposed methods. Experimental results demonstrate the effectiveness of the proposed methods.

Pros:
1. This paper studies an important topic for route planning and specifically focuses on the small subgraph extraction stage.
2. The paper is well written with clear clarity and thus easy to follow.
3. The experimental results generally demonstrate the effectiveness of the proposed methods.


Cons:
1. A notable drawback of the paper lies in the absence of runtime cost evaluation. It is recommended that the authors address this concern and, if possible, supplement the discussion with a summarization of the time complexity analysis for the proposed approximation algorithms.

**Questions:**

1. Please see the above-discussed drawbacks.
2. In addition, can authors also provide the error bound for the approximation algorithms?

**Reviewer Confidence:**

3: The reviewer is confident but not certain that the evaluation is correct

**Scope:**

3: The work is somewhat relevant to the Web and to the track, and is of narrow interest to a sub-community

---

### Official Review · Reviewer_gGTP · 2023-11-29

**Novelty:** 4
**Technical Quality:** 4

**Review:**

The paper proposes an approximation algorithm for subgraph extraction. The authors also prove that the problem is NP-hard.

Although the work looks technically sound, there are several issues.

First and foremost, the main goal is to reduce computational cost of online algorithms on large graph by running them only one subgraph. However, the subgraph extraction is objective driven and when the objective changes the system is going to lose useful information.

Second many graph processing algorithm (for example shortest path) does not need to process full graph, even if it operates on the full graph. The algorithm can on the fly decide which part it wants to explore thereby limiting the utility of one  more lossy preprocessing step like the authors are doing.

**Questions:**

The subgraph is objective driven. What happens the application requires a new objective to be optimized?

Consider shortest path case: how much accuracy the proposed algorithm is going to sacrifice due to its approximation?

For some objective there could be optimal solution and for some there may not be. How can the proposed algorithm guarantee the optimal solution for such objective functions?

**Ethics Review Description:**

None.

**Reviewer Confidence:**

3: The reviewer is confident but not certain that the evaluation is correct

**Scope:**

4: The work is relevant to the Web and to the track, and is of broad interest to the community

---

### Decision · Program_Chairs · 2024-01-22

**Decision:**

Accept

**Comment:**

Summary: The paper looks into algorithm for subgraph extraction for road networks, proving the problem is NP-hard and proposing an approximation.

 Strengths:
 + interesting problem
 + solid theoretical results
 + well-written

 Weaknesses:
 - concern about relevance to the Web
 - lack of runtime cost evaluation

 Recommendation: Accept. Well-executed paper. Web relevance can be further accentuated.